# ON THE LIMITATIONS OF GENERAL PURPOSE DOMAIN GENERALISATION METHODS

## ABSTRACT

The Domain Generalisation (DG) problem setting requires a model trained on a set of data distributions (domains) to generalise to new distributions. Despite a huge amount of empirical study, previous DG methods fail to substantially outperform empirical risk minimisation on rigorous DG benchmarks. Motivated by this, we analyse the DG problem from a learning theoretic perspective and *characterise in which situations DG will succeed or fail*. Specifically, we derive upper bounds on the excess risk of ERM and lower bounds on the minimax excess risk, for three settings with different restrictions on how the domains may differ. In the most unconstrained setting, we show that all learning algorithms converge slowly with respect to number of training domains, potentially explaining the lack of algorithmic progress in this area. We also consider constrained settings including limiting the pairwise domain distances as measured by a broad class of integral probability metrics, and constraining all domains to have the same underlying support. In these constrained cases, DG algorithms can converge more rapidly. Notably, for all three settings, the we demonstrate that ERM has an optimal rate of convergence towards the best possible model. Our analysis guides practitioners interested in knowing when cross-domain generalisation might be reliable, and suggests strategies for optimising the performance of ERM in each setting.

## 1 INTRODUCTION

Machine learning systems have shown exceptional performance on numerous tasks in computer vision, natural language processing, and beyond. However performance drops rapidly when the standard assumption of identically distributed training and testing data is violated. This domain-shift phenomenon occurs widely in many applications of machine learning (Csurka, 2017; Zhou et al., 2021; Koh et al., 2021), and often leads to disappointing results in practical machine learning deployments, since data "in the wild" is almost inevitably different from training sets. Given the practical significance of this issue, numerous methods have been proposed that aim to improve models' robustness to deployment under train-test domain shift by leveraging several different training sets for the same task (Zhou et al., 2021), a problem setting known as Domain Generalisation (DG). These span diverse approaches including specialised neural architectures, data augmentation strategies, and regularisers. Nevertheless, determining the effectiveness of these methods has proven to be difficult. A large scale experimental investigation (Gulrajani & Lopez-Paz, 2021) determined that none of the previously proposed methods for addressing the DG problem could reliably outperform Empirical Risk Minimisation (ERM)—the method that simply ignores all of the domain structure in the data and trains a model as normal.

Meanwhile various theoretical analyses have tried to study the DG problem via the derivation of generalisation bounds, attempting to provide guarantees on the worst case performance of various learning methods in this setting. Usually these analyses make strong assumptions about either the structure of the model or the underlying data generation process. Algorithms based on kernel methods have received some attention (Muandet et al., 2013; Blanchard et al., 2021), and the behaviour of how well these approaches scale with the number of training examples and training domains is well-understood. There is also substantial work on investigating generalisation in other distribution shift settings (Mansour et al., 2009a; Kpotufe, 2017; Rosenfeld et al., 2021a). However, there is a lack of work on exploring the theoretical properties of sensible baselines the fundamental limits of the problem setting.

In this paper we take a different direction to this existing body of theoretical work, and present the first learning-theoretic study of: (i) excess risk bounds for ERM in the DG setting; and (ii) *minimax* excess risk bounds that determine the fundamental limit of the best possible algorithm for a particular setting. To help characterize in what circumstances DG is likely to succeed or fail, we study these bounds for both the most popular and challenging unconstrained DG setting, as well as two less common simpler settings with constraints on how different the domains can be. We show that in the most general unconstrained case, the best possible algorithm converges slowly with respect to the number of domains; while it possible to converge more rapidly in constrained settings where we can guarantee the domains' similarity in terms of pairwise distances, or sharing the same support. Nevertheless, in all cases the minimax bounds and the worst case bounds for ERM have the same scaling behaviour, indicating surprisingly that *it is not possible for any learning algorithm to do substantially better than ERM*. Moreover, we show that out-of-domain generalisation is governed by a similar trade-off between empirical risk and complexity that governs in-domain generalisation. Thus, new architectures or regularisers that improve practical in-domain generalisation of ERM will also improve out-of-domain generalisation, and vice-versa. These results may go some way toward explaining why empirical analyses (Gulrajani & Lopez-Paz, 2021) have shown that the plethora of purpose designed methods (Zhou et al., 2021) struggle to beat ERM when carefully evaluated. Our analysis shows that the search for an alternative learning algorithm to ERM per-se is futile. However, it does provide some specific actionable insights for improving DG by: (1) Regularising more strongly when optimising for out-of-domain generalisation, than would be optimal for in-domain generalisation, and (2) using prior knowledge to develop hypothesis classes that exhibit increased domain invariance and thus move more toward the IPM-constrained variant of the problem.

In summary, we make the following contributions to knowledge:

- We derive upper bounds on the excess risk of ERM that show, as the quantity and variety of training data increases, ERM is able to better approximate the optimal model in the chosen class of models. This conclusion holds for all three of the DG settings that we consider.
- Lower bounds on the minimax excess risk are provided. By comparing these lower bounds with the upper bounds for ERM, we demonstrate that no method can perform substantially better than ERM in the three general purpose DG settings considered in this paper.
- Comparing the upper and minimax bounds across problem settings, we delineate some settings where DG is likely to succeed or fail, in terms of achieving a a reasonable expected error on held out domains after seeing a reasonable number of training domains.
- Actionable insights are given for how one can optimise the performance of ERM in each of the DG settings via the choice of regularisation strength and ensuring the hypothesis class is well-suited to the underlying problem.

## 2 RELATED WORK

The DG problem setting was first introduced and analysed by Blanchard et al. (2011), who also provided a learning theory motivated algorithm for addressing it based on kernel methods. Since then, several other works have also gone down the route of using statistical learning theory to derive kernel methods with performance guarantees for DG (Muandet et al., 2013; Blanchard et al., 2021). We note that a more general analysis of the performance of standard baselines and of fundamental limitations of methods developed for the standard DG setting is missing. However, some work has been undertaken for a variant where unseen domains are restricted to convex or affine combinations of those domains seen during training Rosenfeld et al. (2021a), where it is shown that ERM is minimax optimal. In any case, empirical evidence suggests that these limitations might be closer than we would like. Gulrajani & Lopez-Paz (2021) compared several state of the art methods using DomainBed, a common benchmark and hyper-parameter tuning protocol. They ultimately defend Empirical Risk Minimization (ERM) over more sophisticated alternatives on the grounds that no competitor consistently beats it. We also broadly defend ERM, and go a step further in claiming that one cannot construct a general purpose DG method that substantially outperforms ERM. However, we provide a deeper analysis into when and why ERM works, rooted in a theoretical analysis of generalisation qualities of ERM, unlike the prior purely empirical evaluation.

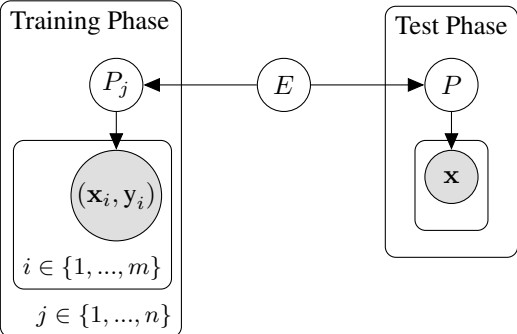

Figure 1: A plate diagram describing an overview of the Domain Generalisation data generation process. The environment—a distribution over domains—is represented by $E$. During training, $n$ domains, $P_1$ to $P_n$, are sampled from $E$. From each of these domains we sample $m$ training points, leading to a total of $mn$ training points. At test time, the model will encounter an indeterminate number of domains, so we do not index the $P$ nodes. From each of these domains, unlabelled points are sampled and the model is required to produce an estimate of the corresponding label.

Methods based on Invariant Causal Prediction (ICP) (Peters et al., 2016) have also been developed for the DG setting, and often come with claims of guaranteed robustness in the face of distribution shifts due to leveraging the underlying causal structure of a problem. The most popular idea in this area is Invariant Risk Minimisation (IRM) (Arjovsky et al., 2019), and variants such as Krueger et al. (2020), which use an objective function inspired by ICP to train a deep neural network for DG. Some other prior work makes a similar observation to us, identifying that standard regularisation methods can improve robustness to distribution shift, albeit through a different lens (Janzing, 2019; Sagawa et al., 2020). However, methods based on IRM have been shown to be more fragile than expected in the face of real distribution shifts, rather than those considered in simplified causal diagrams (Gulrajani & Lopez-Paz, 2021; Rosenfeld et al., 2021b).

Our work also takes inspiration from other transfer learning settings. In particular, there has been a lot of work analysing the supervised and unsupervised Domain Adaptation problems, where there are now a number of ways to relate the performance on training data from some source domain to new data drawn from some target domain (Mansour et al., 2009a;b; Ben-David et al., 2010; Kpotufe, 2017; Zhang et al., 2019; Kpotufe & Martinet, 2020). The focus of these works is typically to define some notion of distribution divergence and then develop a tractable method for estimating this divergence, or finding a model that best exploits the insights provided by such divergences. There have been some attempts to apply these ideas in the DG setting (e.g., Albuquerque et al. (2020)). However, in the domain adaptation settings, there is an emphasis on finding divergences that are capable of exploiting asymmetries in the direction of transfer (Kpotufe & Martinet, 2020). In contrast, the standard DG formulation assumes a domain is just as likely to appear at training time as it is at test time, so it is not clear that we should expect an asymmetric divergence to be well-suited to DG.

## 3 DOMAIN GENERALISATION

**Standard i.i.d. learning** To begin with, we establish some notation by introducing the standard independently and identically distributed (i.i.d.) learning setting. The risk of a model, $f \in \mathcal{F}$, on some distribution, $P$, for some loss, $\ell$, is defined as

$$L_P(f) = \mathop{\mathbb{E}}_{(\mathbf{x},\mathbf{y}) \sim P} [\ell(f(\mathbf{x}), \mathbf{y})].\tag{1}$$

Examples of useful loss functions include the zero–one loss and ramp loss,

$$\ell_{01}(\hat{y}, y) = \mathbf{1}(\text{sign}(\hat{y}) = y), \qquad \ell_{ramp}(\hat{y}, y) = \max(0, \min(1, 1 - \hat{y}y)).\tag{2}$$

Note that the ramp loss is Lipschitz and upper bounds the zero–one loss. Similar to the risk, we can define the empirical risk on an i.i.d. sample, $S_m = \{(\mathbf{x}_i, \mathbf{y}_i)\}_{i=1}^m$, from $P$ as

$$\hat{L}_{S_m}(f) = \frac{1}{m} \sum_{i=1} \ell(f(\mathbf{x}_i), \mathbf{y}_i). \tag{3}$$

Throughout this paper we use Rademacher complexity $\mathcal{R}$, and its empirical analogue $\hat{\mathcal{R}}$, to characterise the capacity of the hypothesis class, $\mathcal{F}$,

$$\mathcal{R}_{P^m}(\mathcal{F}) = \mathbb{E}_{S_m}[\hat{\mathcal{R}}_{S_m}(\mathcal{F})], \qquad \hat{\mathcal{R}}_{S_m}(\mathcal{F}) = \mathbb{E}_{\sigma}\left[\sup_{f \in \mathcal{F}} \frac{1}{m} \sum_{i=1}^m \sigma_i f(\mathbf{x}_i)\right], \tag{4}$$

where $\sigma$ is a vector Rademacher random variables, so $\Pr(\sigma_i = 1) = \Pr(\sigma_i = -1) = \frac{1}{2}$. In the case of linear models, these Rademacher complexity can be controlled by constraints on weight norms, while in the case of deep models they further depend on properties of the chosen network architecture. For typical choices of hypothesis class, the complexity scales as $\mathcal{O}(1/\sqrt{m})$. For simplicity, we avoid introducing multi-output generalisations of Rademacher complexity and therefore focus only on binary classification, with the sign of the model output indicating the predicted class. A typical generalisation bound using Rademacher complexity in the i.i.d. setting has the form given in the theorem below.

**Theorem 1** (Mohri et al. (2018)). *Suppose $\ell$ takes values in $[0, 1]$ and $S_m$ is contains $m$ i.i.d. samples from $P$. The worst-case difference between the population risk and empirical risk for models selected from $\mathcal{F}$ is bounded, in expectation, by*

$$\mathbb{E}_{S_m}\left[\sup_{f \in \mathcal{F}} L_P(f) - \hat{L}_{S_m}(f)\right] \leq 2\mathcal{R}_{P^m}(\ell \circ \mathcal{F}),$$

*and with probability at least $1 - \delta$ over realisations of $S_m$, we have for all $f \in \mathcal{F}$ that*

$$L_P(f) \leq \hat{L}_{S_m}(f) + 2\hat{\mathcal{R}}_{S_m}(\ell \circ \mathcal{F}) + 3\sqrt{\frac{\ln 2/\delta}{2m}}.$$

*where we have used $\ell \circ \mathcal{F} = \{(\boldsymbol{x}, y) \mapsto \ell(f(\boldsymbol{x}), y) \,:\, f \in \mathcal{F}\}$.*

This is a variation of a classic result attributed to Bartlett & Mendelson (2002).

**From i.i.d. to DG learning** In the Domain Generalisation setting we are interested in two layer hierarchical data generating processes, where the top level distribution is denoted by $E$. One can sample domain distributions from $E$, and subsequently sample data points from each domain distribution. The most common way to formulate the DG problem setting assumes that one has access to training data from $n$ domain distributions sampled i.i.d. from $E$. For ease of exposition we assume that we sample $m$ data points from each domain, but we note that similar versions of our results still hold in the case where a different number of data points are available for each domain. The task is then to build a model that can still perform well when applied to data drawn from novel domain distributions sampled from $E$ at test time. The DG data generation process is summarised by the plate diagram in Figure 1. Just as the risk is the central quantity of interest in the i.i.d. setting, in the DG setting we are concerned with the transfer risk,

$$L_E(f) = \mathbb{E}_{P \sim E}[L_P(f)], \tag{5}$$

and the corresponding empirical risk is given by

$$\hat{L}_{S_{mn}}(f) = \frac{1}{n} \sum_{j=1}^n \hat{L}_{S_m^j}(f), \tag{6}$$

where $S_m^j$ is the training data sampled from $P_j$ and $S_{mn} = \cup_{j=1}^n S_m^j$ is a set containing all the training data. One of our objectives is proving generalisation bounds along the same lines as Theorem 1. To do this, we extend the idea of Rademacher complexity to be defined for two-level distributions, $E$,

$$\mathcal{R}_{E^n}(\mathcal{F}) = \mathbb{E}_{P_{1:n}} \mathbb{E}_{\sigma}\left[\sup_{f \in \mathcal{F}} \frac{1}{n} \sum_{j=1}^n \sigma_j \mathbb{E}_{\mathbf{x} \sim P_j}[f(\mathbf{x})]\right]. \tag{7}$$

Two models that feature prominently in our analysis include the Empirical Risk Minimiser within the class of models, $\mathcal{F}$,

$$\hat{f} = \arg\min_{f \in \mathcal{F}} \hat{L}_{S_{mn}}(f), \tag{8}$$

and the optimal model in $\mathcal{F}$,

$$f^* = \arg\min_{f \in \mathcal{F}} L_E(f). \tag{9}$$

The difference in risk between a model built using some learning algorithm (e.g., ERM) and the optimal model, $f^*$, is referred to as the excess risk. Central to our analysis is the *minimax* excess risk for learning algorithms, $A$, that map from training sets to elements of $\mathcal{F}$. In the i.i.d. setting this is given by

$$M(\mathcal{F}, \mathcal{P}, m) = \inf_{A} \sup_{P \in \mathcal{P}} \left\{ \mathop{\mathbb{E}}_{S_m} \left[ L_P(A(S_m)) \right] - L_P(f^*) \right\}, \tag{10}$$

where $\mathcal{P}$ is the set of all i.i.d. distributions over the cartesian product of a feature space and label space, $X \times Y$. This definition can be straightforwardly extended to the DG setting by considering a set of distributions other than $\mathcal{P}$. For the remainder of this paper we consider various subsets of $\mathcal{E}$, the set of all possible distributions that follow the two layer structure encountered in DG. With some abuse of notation, we will denote the minimax excess risk for selecting models from $\mathcal{F}$ for distributions in $\mathcal{E}$ using $m$ training examples from each of $n$ domains by $M(\mathcal{F}, \mathcal{E}, mn)$.

The minimax excess risk is a lower bound on the best possible excess risk of any algorithm when applied to the hardest problem in $\mathcal{E}$. It can be thought of in a game-theoretic way: one player constructs an algorithm to minimise the excess risk; meanwhile, the other player selects a pathological distribution for this algorithm to maximise the excess risk. It provides an indication of how quickly the best possible learning algorithm is able to find the optimal model as a function of the amount of training data (and number of training domains) available. One could argue that $\mathcal{E}$ contains many DG problems that are not likely to be of interest in the real world, and that one of the uninteresting problems could be the maxmimiser in the minimax excess risk. In the remainder of the paper we consider three different restricted subsets of $\mathcal{E}$ that correspond to popular intuitions discussed in the transfer learning literature: the general case; problems with bounded pairwise distance between domains; and problems where the marginals for each domain have bounded density ratios.

## 4 WHEN IS DG FEASIBLE?

We derive upper bounds on the excess risk of ERM in each setting, and lower bounds on the minimax excess risk for each setting. By looking at the bounds in each setting, and the gap between them, we aim to understand the relative difficulty of each scenario in terms of number of domains required for learning, and whether one can improve on the baseline ERM algorithm.

### 4.1 THE GENERAL SETTING

This is the setting most commonly considered in the literature. In this setting the DG problem can be any $E \in \mathcal{E}$, a hierarchical distribution with a structure as described in Figure 1.

#### 4.1.1 THE EXCESS RISK OF ERM

To begin with, we provide a uniform convergence result for the generalisation error of methods developed for this setting. This theorem is used in the course of proving an upper bound on the excess risk of ERM, but is also interesting in its own right, as it applies to all DG algorithms developed for the general setting.

**Theorem 2.** *Assume $\ell$ takes values in $[0, 1]$. For a hypothesis class, $\mathcal{F}$, and any $E \in \mathcal{E}$, we have that*

$$\mathop{\mathbb{E}}_{S_{mn}} \left[ \sup_{f \in \mathcal{F}} L_E(f) - \hat{L}_{S_{mn}}(f) \right] \leq 2 \mathop{\mathbb{E}}_{P_{1:n}} \left[ \mathcal{R}_{P_{1:n}^m}(\ell \circ \mathcal{F}) \right] + 2\mathcal{R}_{E^n}(\ell \circ \mathcal{F}).$$

*Moreover, we have with confidence at least $1 - \delta$ over the realisations of $S_{mn}$, for all $f \in \mathcal{F}$, that*

$$L_E(f) \leq \hat{L}_{S_{mn}}(f) + 2\hat{\mathcal{R}}_{S_{mn}}(\ell \circ \mathcal{F}) + \frac{2}{m} \sum_{i=1}^{m} \hat{\mathcal{R}}_{S_n^i}(\ell \circ \mathcal{F}) + 5\sqrt{\frac{\ln 3/\delta}{2mn}},$$

*where we denote by $S_n^i$ the set containing the the $i$th example from each $S_m^j$.*

The proof can be found in Appendix B.

**Discussion** Theorem 2 tells us that the expected gap between the transfer risk and empirical risk is bounded by two Rademacher complexity terms. The second part of the result provides an observable version of this bound that holds with high probability and has the same scaling behaviour, but is slightly looser due to the additional sampling error term. For typical hypothesis classes, the first term controls how much the model could have overfit to the specific points sampled in the training set and will scale as $\mathcal{O}(1/\sqrt{mn})$. The second term controls how much the model could have overfit to the domains included in the training set, and will scale as $\mathcal{O}(1/\sqrt{n})$. Mirroring conventional generalisation in standard i.i.d. learning, a very simple model may minimise the Rademacher terms while producing high empirical risk and vice-versa. Thus, good generalisation critically depends on a carefully chosen empirical risk versus model complexity trade-off.

We next bound the excess risk between the ERM solution, $\hat{f}$, and the best possible model, $f^*$, within the function class $\mathcal{F}$.

**Theorem 3.** *Under the same conditions as Theorem 2, the excess risk of ERM is bounded as*

$$\mathbb{E}_{S_{mn}}\left[L_E(\hat{f})\right] - L_E(f^*) \leq 2 \mathbb{E}_{P_{1:n}}\left[\mathcal{R}_{P_{1:n}^m}(\ell \circ \mathcal{F})\right] + 2\mathcal{R}_{E^n}(\ell \circ \mathcal{F}).$$

The proof can be found in Appendix C

**Discussion** Theorem 3 tells us that the gap between ERM and the *best possible* predictor in the function class depends on the same complexity terms observed in Theorem 2. This implies that, for any typical hypothesis class, ERM will find a near-optimal model if given sufficient data. In particular, the rate of convergences towards the optimal model will be $\mathcal{O}(1/\sqrt{mn} + 1/\sqrt{n})$, indicating that it is not only the total volume of training data that is important, but also the number of training domains.

### 4.1.2 A Lower Bound on the Minimax Excess Risk

We now turn to the problem of computing lower bounds on the minimax excess risk for general DG problems. We have already provided an upper bound on the excess risk for using ERM to select a model from $\mathcal{F}$ which holds for all problems in $\mathcal{E}$. The goal of deriving the lower bound on the minimax risk, given below, is to determine how much room there is to improve upon ERM.

**Theorem 4.** *Let $\mathcal{F}$ be a class of models producing predictions in $\{-1, 1\}$ and let $\ell$ be the zero–one loss. We have that the minimax excess risk for problems in $\mathcal{E}$ is bounded from below by*

$$M(\mathcal{F}, \mathcal{E}, mn) \geq \frac{\sup_{S_{2mn}} \hat{\mathcal{R}}_{S_{2mn}}(\mathcal{F}) + \sup_{S_{2n}} \hat{\mathcal{R}}_{S_{2n}}(\mathcal{F})}{4} - \frac{\sup_{S_{mn}} \hat{\mathcal{R}}_{S_{mn}}(\mathcal{F}) + \sup_{S_n} \hat{\mathcal{R}}_{S_n}(\mathcal{F})}{8}.$$

The proof is in Appendix D.

**Discussion** The results in Theorems 3 and 4 tell us that the gap between $f^*$ and a model trained by the best possible learning algorithm can be lower bounded by the same types of complexity terms used in the upper bound for ERM. For hypothesis classes of interest in machine learning we can say that the minimax excess risk scales as

$$\Omega\left(\frac{1}{\sqrt{mn}} + \frac{1}{\sqrt{n}}\right) - \Omega\left(\frac{1}{\sqrt{mn}} + \frac{1}{\sqrt{n}}\right) = \Omega\left(\frac{1}{\sqrt{mn}} + \frac{1}{\sqrt{n}}\right). \tag{11}$$

Moreover, this best-case behaviour matches the worst-case scaling behaviour of ERM. As such, we can conclude that *ERM already has the optimal scaling behaviour for general DG problems.* This reasoning about asymptotic rates can obfuscate the impact of the constant factors, which becomes more important for smaller $n$. We note that the upper and lower bounds: (i) have relatively close constant factors; and (ii) the same function (i.e., Rademacher complexity) governs both in-domain generalisation and cross-domain generalisation, so the number of training examples are needed for generalisation in the i.i.d. setting will be indicative of the number of training domains needed to have good generalisation in the DG setting.

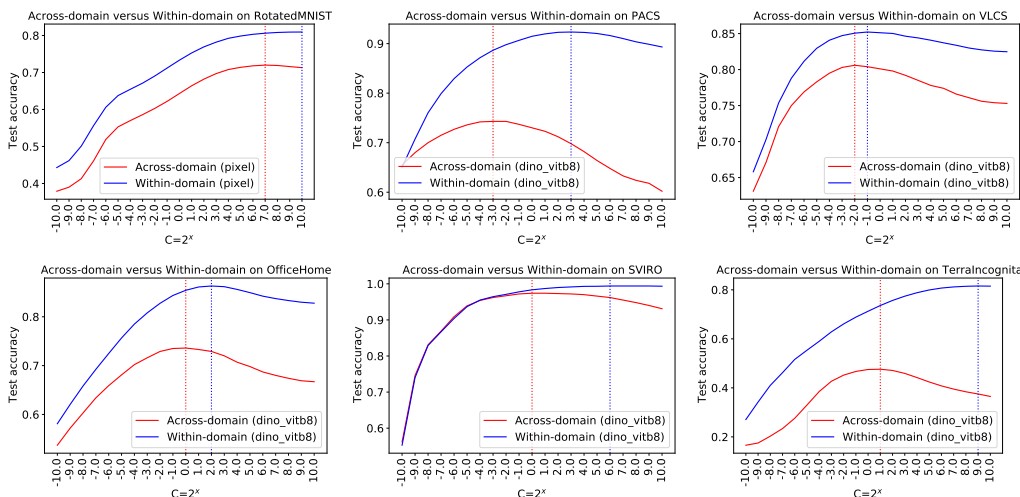

Figure 2: Linear SVM performance on DomainBed benchmark datasets. Optimal tuning for performance on novel target domains (red) always requires stronger regularisation (lower $C$) than for performance on seen domains (blue).

**Practical Insights**  While the results so far appear to be pessimistic, there are actionable insights that we discuss here. The difference between Theorem 2 and single domain bounds (e.g., Theorem 1) is the additional dependence on the number of domains $n$ via the additional Rademacher complexity term. This shows that in-domain generalisation and cross-domain generalisation are both controlled by the same type of complexity term. This means that novel architectures or in-domain regularisation methods (interventions that reduce complexity while maintaining the empirical risk, or vice-versa) will automatically benefit cross-domain generalisation: Seeking models with good in-domain generalisation will also improve DG. This reflects the situation in empirical DG benchmarks where underlying neural architecture improvement has the most substantial influence on DG performance (Gulrajani & Lopez-Paz, 2021; Zhou et al., 2021), as well as in-domain performance.

More specifically, comparing the theorems also shows that when the goal is to generalise to new domains, the risk of overfitting is higher. *Therefore a lower complexity model is better for held out domain performance compared for seen domain performance in standard i.i.d. learning.* To illustrate this empirically, we train linear SVM on pre-computed DINO (Caron et al., 2021) deep features[1] for six popular DG benchmarks. Tight bounds on linear SVM model complexity are known and can be directly controlled by the scalar SVM slack parameter. The objective is also convex so confounding factors in deep network training (stochastic optimisers, early stopping, etc) disappear, and training is fast so that we can densely and exhaustively evaluate a wide range of complexities. We compare conditions corresponding to the two theorems: (i) Training on the train splits of all domains, and testing on the test splits of all domains (standard i.i.d. learning). (ii) Training on the train splits of three domains, and testing on the test splits of a held out domain (i.e., DG).

The results in Fig. 2 average over 5 random seeds for dataset splitting, and all choices of held-out target domain. From these we can see that: (i) All experiments exhibit the classic trade-off between fitting the data well and constraining hypothesis class complexity appropriately. There is underfitting for high regularisation (small C), and overfitting at low regularisation (large C). (ii) Across-domain/DG condition (red) exhibits lower performance as expected due due to the distribution shift. And most importantly (iii) the optimal regularisation for novel-domain performance is stronger than for seen-domain performance (red vertical lines left of blue). This illustrates our theoretical result that the ideal model complexity is lower for DG than for conventional i.i.d. learning.

---

[1]Note that using a *fixed* feature extractor trained on independent data does not impact the model complexity or associated generalisation bound.

## 4.2 THE BOUNDED INTEGRAL PROBABILITY METRIC SETTING

The general setting analysed in Section 4.1 allows for arbitrarily different marginal distributions, $P(\mathbf{x})$ and $P'(\mathbf{x})$, across two different domains, $P$ and $P'$. We next analyse whether making assumptions on their similarity could lead to stronger guarantees, and discuss whether this restriction enables algorithms to improve on ERM.

Integral Probability Metrics are a broad class of metrics for measuring distances between probability distributions. The generalised definition of an IPM is

$$d_{\mathcal{G}}(P, P') = \sup_{g \in \mathcal{G}} \left| \mathbb{E}_{\mathbf{z} \sim P}[g(\mathbf{z})] - \mathbb{E}_{\mathbf{z} \sim P'}[g(\mathbf{z})] \right|, \tag{12}$$

where different choices of $\mathcal{G}$ lead to different metrics, and in our case $\mathbf{z}$ will represent the tuple $(\mathbf{x}, \mathbf{y})$. For various choices of $\mathcal{G}$ one can recover distribution distances commonly found in the DG and domain adaptation literature, such as maximum mean discrepancy, $d_{H \triangle H}$-distance, discrepancy distance, total variation distance, and more. In this section we analyse a subset of DG problems $\mathcal{E}_{\mathcal{G},\sigma}$, where the distance between different domains, as measured by some IPM $d_{\mathcal{G}}$, is bounded by $\sigma$,

$$\mathcal{E}_{\mathcal{G},\sigma} = \{E \mid E \in \mathcal{E} \land \forall P, P' \in \operatorname{supp} E, d_{\mathcal{G}}(P, P') \leq \sigma\}. \tag{13}$$

As in the general setting, we start with a uniform convergence result that will be useful in the subsequent analysis.

**Theorem 5.** *Let $\ell$ be a loss function taking values in $[0, 1]$. For a hypothesis class, $\mathcal{F}$, and any $\mathcal{G}$ such that $\ell \circ \mathcal{F} \subseteq \mathcal{G}$, then we have for any $E \in \mathcal{E}_{\mathcal{G},\sigma}$*

$$\mathbb{E}_{S_{mn}} \left[ \sup_{f \in \mathcal{F}} L_E(f) - \hat{L}_{S_{mn}}(f) \right] \leq 2 \mathbb{E}_{P_{1:n}} [\mathcal{R}_{P_{1:n}^m}(\ell \circ \mathcal{F})] + \sigma.$$

*Moreover, we have with confidence at least $1 - \delta$ over the realisations of $S_{mn}$, for all $f \in \mathcal{F}$, that*

$$L_E(f) \leq \hat{L}_{S_{mn}}(f) + 2\hat{\mathcal{R}}_{S_{mn}}(\ell \circ \mathcal{F}) + \sigma + 3\sqrt{\frac{\ln 2/\delta}{2mn}}.$$

The proof is given in Appendix E. Similar to the more general setting, we also obtain a bound on the excess risk of ERM.

**Corollary 1.** *Under the same conditions as Theorem 5, the excess risk of ERM is bounded by*

$$\mathbb{E}_{S_{mn}} [L_E(\hat{f})] - L_E(f^*) \leq 2 \mathbb{E}_{P_{1:n}} [\mathcal{R}_{P_{1:n}^m}(\ell \circ \mathcal{F})] + \sigma.$$

The proof is essentially the same as for Theorem 3, except we apply Theorem 5 instead of Theorem 2.

**Discussion** This theorem tells us that if all domains associated with the DG problem exhibit a high degree of similarity, as measured by a sufficiently expressive IPM, the effect of the distribution shift is reduced and one mainly needs to consider the overfitting behaviour typically seen in i.i.d. problems. This result is of most interest when only a small number of training domains are available; if $\sigma$ is small, one can still guarantee that not too much additional error will be incurred in the DG setting, despite having seen little variety in training domains. It is also worth noting that $\mathcal{E}_{\mathcal{G},\sigma} \subseteq \mathcal{E}$, so the guarantee from Theorem 3 still applies. This means that one can still expect the transfer risk to improve as more training domains are collected, and the large-$n$ scaling behaviour of $\mathcal{O}(1/\sqrt{mn} + 1/\sqrt{n})$ is carried over.

**Theorem 6.** *Let $\mathcal{F}$ be a class of models producing predictions in $\{-1, 1\}$ and $\ell$ be the zero–one loss. For any $\mathcal{G}$ such that $d_{\mathcal{G}}$ takes values in $[0, 1]$, we have that the minimax excess risk for problems in $\mathcal{E}_{\mathcal{G},\sigma}$ is bounded from below by*

$$M(\mathcal{F}, \mathcal{E}_{\mathcal{G},\sigma}, mn) \geq \frac{\sup_{S_{2mn}} \hat{\mathcal{R}}_{S_{2mn}}(\mathcal{F}) + \sup_{S_{2n}} \sigma \hat{\mathcal{R}}_{S_{2n}}(\mathcal{F})}{4} - \frac{\sup_{S_{mn}} \hat{\mathcal{R}}_{S_{mn}}(\mathcal{F}) + \sup_{S_n} \sigma \hat{\mathcal{R}}_{S_n}(\mathcal{F})}{8}.$$

The proof is given in Appendix F.

**Discussion** Because we have two upper bounds on the excess risk of ERM (Theorem 3 and Corollary 1), it is easiest to compare the upper and lower bounds separately for the large-$n$ and small-$n$ cases. In the large-$n$ case, the ERM excess risk and this lower bound behave in the same way as Theorems 3 and 4, respectively, so we can conclude that in the large-$n$ case there is once again no substantial room for improvement over ERM. In the small-$n$ case, $\sup_{S_n} \hat{\mathcal{R}}_{S_n}(\mathcal{F})$ and $\sup_{S_{2n}} \hat{\mathcal{R}}_{S_{2n}}(\mathcal{F})$ will both be close to one, so the corresponding terms that they appear in will be approximately equal to $\sigma$. This implies that in the small-$n$ case, the lower bound in Theorem 6 will be close to the upper bound in Corollary 1, so we can again conclude that there is little room for improvement over ERM.

### 4.2.1 WHEN CAN $\sigma$ BE SMALL?

Given the full generality of IPMs, it may not be obvious what kinds of DG problems are likely to have small bounds on the IPM between their domains. In this section we provide two examples of cases where one can expect a small value for $\sigma$.

**Domains are mixtures with common components**  If every domain $P \in \operatorname{supp} E$ can be written as a mixture over a fixed set of components, $Q_{1:k}$, but with random mixture coefficients, $\alpha_{1:k}$,

$$P = \sum_{c=1}^{k} \alpha_c Q_c, \qquad \sum_{c=1}^{k} \alpha_{j,c} = 1, \tag{14}$$

then the IPM with $TV$, the set of all functions mapping to $[0, 1]$, between two domains is bounded by the sum of absolute differences in the corresponding $\alpha$ vectors, so

$$d_{TV}(P, P') \leq \frac{1}{2} \sum_{c=1}^{k} |\alpha_c - \alpha'_c|. \tag{15}$$

This follows because $d_{TV}$ is the total variation distance, which can also be expressed as half the $L^1$ distance between the densities. If there are non-trivial bounds on the support of the mixing coefficients then Equation 15 will be less than one and we will obtain a non-vacuous $\sigma$. An example of where this can appear in real-world setting is in the crossover between DG and fairness, where different domains can correspond to different mixtures of demographics between populations.

**Invariant Hypothesis Classes**  Let $G$ be a group that acts on elements of $X$. Suppose that for all $P, P' \in \operatorname{supp} E$ there exists $g \in G$ such that $P(\mathbf{x}, \mathbf{y}) = P'(g\mathbf{x}, \mathbf{y})$. A hypothesis class can be said to be $(\epsilon, G)$-invariant w.r.t. $E$ if for $(\mathbf{x}, \mathbf{y}) \sim P$

$$\left| \mathbb{E}_{(\mathbf{x}, \mathbf{y})} [\ell(f(\mathbf{x}), \mathbf{y}) - \ell(f(g\mathbf{x}), \mathbf{y})] \right| \leq \epsilon, \quad \forall g \in G, f \in \mathcal{F}. \tag{16}$$

From this definition, it is clear that when $\mathcal{F}$ is $(\epsilon, G)$-invariant the IPM defined via $\ell \circ \mathcal{F}$, for any two domains $P, P' \in \operatorname{supp} E$, the IPM is bounded as

$$d_{\ell \circ \mathcal{F}}(P, P') \leq \epsilon. \tag{17}$$

This style of invariance is commonly discussed in the geometric deep learning literature Bronstein et al. (2021), where this is a lot of work on developing hypothesis classes with specific group invariance properties. For example, in the design of machine learning models that operate on molecules to enable drug discovery pipelines (Igashov et al., 2024). Leveraging models with invariance properties also common in computer vision; we provide an empirical demonstration of how $(G, \epsilon)$-invariance can lead to good transfer risk in the image recognition context in Appendix I.

Note that this invariance property must hold for every model in the hypothesis class. One cannot use the training data, $S_{mn}$, to attempt to identify a subset of models that satisfy this condition and obtain this more favourable guarantee. However, there is a growing literature on using auxiliary unlabelled data to learn feature encoders with approximate invariances to various transformations Ericsson et al. (2022); Chavhan et al. (2023).

### 4.3 THE BOUNDED DENSITY RATIO SETTING

The bounded IPM setting analysed in Section 4.2 assume there is some level of similarity between domains, in the sense that some data points could be observed in more than one domain. However,

the possibility of some examples occurring exclusively in a single domain is still left open. In contrast, this Section considers the bounded density ratio setting, where each domain in a DG problem has identical support. This means that if an example can be observed in one domain, then it can also be observed in any other domain, albeit with a different probability. Formally, we define the class of $\rho$-bounded density ratio DG problems, for any $\rho \geq 1$, as a subset of $\mathcal{E}$,

$$\mathcal{E}_\rho = \left\{ E \mid E \in \mathcal{E} \ \wedge \ \forall P, P' \in \mathrm{supp}\, E, \left\| \frac{dP}{dP'} \right\|_\infty \leq \rho \right\}, \tag{18}$$

where we use $dP$ to denote the probability density function of $P$.

**Theorem 7.** *Let $\ell$ be a loss function taking values in $[0,1]$. For a hypothesis class, $\mathcal{F}$, and any $E \in \mathcal{E}_\rho$, we have that*

$$\underset{S_{mn}}{\mathbb{E}} \left[ L_E(\hat{f}) \right] - L_E(f^*) \leq 2\rho \underset{P_{1:n}}{\mathbb{E}} \left[ \mathcal{R}_{P_{1:n}^m}(\mathcal{F}) \right] + (\rho - 1) \underset{S_{mn}}{\mathbb{E}} \left[ \hat{L}_{S_{mn}}(\hat{f}) \right].$$

The proof is given in Appendix G.

**Discussion** One of the key differences between this setting and previous settings is that there is no term that scales as $\mathcal{O}(1/\sqrt{n})$, indicating that one could actually achieve good transfer risk with a small number of training domains. There are two ways to interpret this result in more detail. The first is that if one uses ERM to train an interpolating classifier (i.e., a classifier achieving zero training error) with sufficient data to make the Rademacher complexity term small, then the model will generalise well to new domains. Substantial empirical evidence exists to suggest that deep neural networks often interpolate the training data. We also note that this upper bound does not stipulate that training data must come from a wide variety of training domains; a substantial volume of data from one domain is enough. The second interpretation of this bound is arises from noting that $\mathbb{E}_{S_{mn}}[\hat{L}_{S_{mn}}(\hat{f})] \leq L_E(f^*)$. This implies that if the hypothesis class is "well-specified", in the sense that it contains the ground truth labelling function, then one only needs sufficient data from a single domain to find the best model via ERM.

**Theorem 8.** *Let $\mathcal{F}$ be a class of models producing predictions in $\{-1, 1\}$ and $\ell$ be the zero–one loss. We have that the minimax risk for problem in $\mathcal{E}_\rho$ is bounded from below by*

$$M(\mathcal{F}, \mathcal{E}_\rho, mn) \geq \sup_{S_{2mn}} \frac{1}{2} \hat{\mathcal{R}}_{S_{2mn}}(\mathcal{F}) - \frac{1}{4} \sup_{S_{mn}} \hat{\mathcal{R}}(\mathcal{F}).$$

The proof is given in Appendix H.

**Discussion** Assuming a hypothesis class capable of obtaining low training error is chosen, the minimax excess risk lower bound scales at the same rate as the upper bound on the excess risk of ERM: $\Theta(1/\sqrt{mn})$.

## 5 CONCLUSION

This paper provides a theoretical analysis of several DG problem settings through the use of bounds on the excess risk of ERM and the minimax excess risk. Our results delineate two constrained settings where it is possible to generalise with a smaller number of training domains, as opposed to the most general setting where a large number of training domains are required for learning. However, in all cases it is not possible to substantially outperform ERM. Therefore we also provide several insights into how one can maximise ERM performance in each setting. In the unconstrained setting, the only course of action is to heavily regularise the model, as the model complexity has a larger impact on the transfer risk than the usual i.i.d. risk. If the bounded density ratio assumption holds, it is crucial to specify the "correct" model class—in the sense that the ground truth labelling function is well approximated by a function in the hypothesis class. The bounded IPM setting is unique in that the modeller can influence whether it holds or not by choosing an appropriate hypothesis class. We have provided two examples, along with an experimental demonstration in Appendix I, that carefully choosing a hypothesis class with the correct invariances for a problem can substantially reduce the number of domains required for good DG performance. Going forward, we anticipate that data-driven approaches to defining hypothesis classes via auxiliary data (e.g., through transfer learning or meta-learning) will enable appropriate hypothesis classes to be constructed with minimal manual labour.

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

## A    APPENDIX

## B    PROOF OF THEOREM 2

**Theorem 2.** *Assume $\ell$ takes values in $[0,1]$. For a hypothesis class, $\mathcal{F}$, and any $E \in \mathcal{E}$, we have that*

$$\mathbb{E}_{S_{mn}}\left[\sup_{f\in\mathcal{F}} L_E(f) - \hat{L}_{S_{mn}}(f)\right] \le 2\mathbb{E}_{P_{1:n}}\left[\mathcal{R}_{P_{1:n}^m}(\ell\circ\mathcal{F})\right] + 2\mathcal{R}_{E^n}(\ell\circ\mathcal{F}).$$

*Moreover, we have with confidence at least $1-\delta$ over the realisations of $S_{mn}$, for all $f \in \mathcal{F}$, that*

$$L_E(f) \le \hat{L}_{S_{mn}}(f) + 2\hat{\mathcal{R}}_{S_{mn}}(\ell\circ\mathcal{F}) + \frac{2}{m}\sum_{i=1}^{m}\hat{\mathcal{R}}_{S_n^i}(\ell\circ\mathcal{F}) + 5\sqrt{\frac{\ln 3/\delta}{2mn}},$$

*where we denote by $S_n^i$ the set containing the the ith example from each $S_m^j$.*

*Proof.* We can decompose the DG risk as

$$\mathbb{E}_{S_{mn}}\left[\sup_{f\in\mathcal{F}} L_E(f) - \hat{L}_{S_{mn}}(f)\right] \tag{19}$$

$$= \mathbb{E}_{S_{mn}}\left[\sup_{f\in\mathcal{F}} L_E(f) - \frac{1}{n}\sum_{j=1}^{n} L_{P_j}(f) + \frac{1}{n}\sum_{j=1}^{n} L_{P_j}(f) - \hat{L}_{S_{mn}}(f)\right] \tag{20}$$

$$\le \mathbb{E}_{P_{1:n}}\left[\sup_{f\in\mathcal{F}} L_E(f) - \frac{1}{n}\sum_{j=1}^{n} L_{P_j}(f)\right] + \mathbb{E}_{S_{mn}}\left[\sup_{f\in\mathcal{F}}\frac{1}{n}\sum_{j=1}^{n} L_{P_j}(f) - \hat{L}_{S_{mn}}(f)\right], \tag{21}$$

where the inequality arises from splitting the supremum into two terms. The theorem will follow by providing upper bounds on each of these expectations and plugging them back into this decomposition. The expectations are bounded using small variations of the symmetrisation argument typically used to prove Rademacher complexity-based generalisation bounds.

The first expectation is bounded by

$$
\mathop{\mathbb{E}}_{P_{1:n}} \left[ \sup_{f \in \mathcal{F}} L_E(f) - \frac{1}{n} \sum_{j=1}^{n} L_{P_j}(f) \right] \tag{22}
$$

$$
= \mathop{\mathbb{E}}_{P_{1:n}} \left[ \sup_{f \in \mathcal{F}} \mathop{\mathbb{E}}_{\tilde{P}_{1:n}} \left[ \frac{1}{n} \sum_{j=1}^{n} L_{\tilde{P}_j}(f) \right] - \frac{1}{n} \sum_{j=1}^{n} L_{P_j}(f) \right] \tag{23}
$$

$$
\leq \mathop{\mathbb{E}}_{P_{1:n}} \mathop{\mathbb{E}}_{\tilde{P}_{1:n}} \left[ \sup_{f \in \mathcal{F}} \frac{1}{n} \sum_{j=1}^{n} (L_{\tilde{P}_j}(f) - L_{P_j}(f)) \right] \tag{24}
$$

$$
= \mathop{\mathbb{E}}_{P_{1:n}} \mathop{\mathbb{E}}_{\tilde{P}_{1:n}} \mathop{\mathbb{E}}_{\sigma} \left[ \sup_{f \in \mathcal{F}} \frac{1}{n} \sum_{j=1}^{n} \sigma_j (L_{\tilde{P}_j}(f) - L_{P_j}(f)) \right] \tag{25}
$$

$$
\leq \mathop{\mathbb{E}}_{P_{1:n}} \mathop{\mathbb{E}}_{\sigma} \left[ \sup_{f \in \mathcal{F}} \frac{1}{n} \sum_{j=1}^{n} \sigma_j L_{P_j}(f) \right] + \mathop{\mathbb{E}}_{\tilde{P}_{1:n}} \left[ \sup_{f \in \mathcal{F}} \frac{1}{n} \sum_{j=1}^{n} \sigma_j L_{\tilde{P}_j}(f) \right] \tag{26}
$$

$$
= 2\mathcal{R}_{E^n}(\ell \circ \mathcal{F}), \tag{27}
$$

where the first inequality comes from exchanging the supremum and expectations, and the second inequality comes from the subadditivity of suprema.

The second expectation in the decomposition is bounded via Theorem 1, yielding the first part of the result. It also allows us to conclude

$$
L_E(f) \leq \hat{L}_{S_{mn}}(f) + 2\hat{\mathcal{R}}_{S_{mn}}(\ell \circ \mathcal{F}) + 2\mathcal{R}_{E^n}(\ell \circ \mathcal{F}) + 3\sqrt{\frac{\ln 2/\delta}{2mn}}. \tag{28}
$$

To obtain the second part of the result, we proceed as follows,

$$
\mathcal{R}_{E^n}(\ell \circ \mathcal{F}) = \mathop{\mathbb{E}}_{P_{1:n}} \left[ \sup_{f \in \mathcal{F}} \frac{1}{n} \sum_{j=1}^{n} \sigma_j L_{P_j}(f) \right] \tag{29}
$$

$$
\leq \mathop{\mathbb{E}}_{P_{1:n}} \mathop{\mathbb{E}}_{(\mathbf{x}_{1:n}, \mathbf{y}_{1:n})} \mathop{\mathbb{E}}_{\sigma} \left[ \sup_{f \in \mathcal{F}} \frac{1}{n} \sum_{j=1}^{n} \sigma_j \ell(f(\mathbf{x}_j), \mathbf{y}_j) \right], \tag{30}
$$

which we observe is the expected value of

$$
\frac{1}{m} \sum_{i=1}^{m} \hat{\mathcal{R}}_{S_m^i}(\ell \circ \mathcal{F}). \tag{31}
$$

This satisfies the bounded difference property with constant $\frac{1}{mn}$, so from McDiarmid's inequality, we have for all $f \in \mathcal{F}$, with probability at least $1 - \delta$,

$$
\mathcal{R}_{E^n}(\ell \circ \mathcal{F}) \leq \frac{1}{m} \sum_{i=1}^{m} \hat{\mathcal{R}}_{S_n^i}(\ell \circ \mathcal{F}) + \sqrt{\frac{\ln 1/\delta}{2mn}}. \tag{32}
$$

The second part of the theorem follows from combining this with Equation 28 via the union bound.

$\square$

## C  PROOF OF THEOREM 3

**Theorem 3.** *Under the same conditions as Theorem 2, the excess risk of ERM is bounded as*

$$
\mathop{\mathbb{E}}_{S_{mn}} [L_E(\hat{f})] - L_E(f^*) \leq 2 \mathop{\mathbb{E}}_{P_{1:n}} \left[ \mathcal{R}_{P_{1:n}^m}(\ell \circ \mathcal{F}) \right] + 2\mathcal{R}_{E^n}(\ell \circ \mathcal{F}).
$$

*Proof.*

$$\mathop{\mathbb{E}}_{S_{mn}} \left[ L_E(\hat{f}) \right] - L_E(f^*) \tag{33}$$

$$= \mathop{\mathbb{E}}_{S_{mn}} \left[ L_E(\hat{f}) - \hat{L}_{S_{mn}}(\hat{f}) + \hat{L}_{S_{mn}}(\hat{f}) - \hat{L}_{S_{mn}}(f^*) + \hat{L}_{S_{mn}}(f^*) \right] - L_E(f^*) \tag{34}$$

$$= \mathop{\mathbb{E}}_{S_{mn}} \left[ L_E(\hat{f}) - \hat{L}_{S_{mn}}(\hat{f}) + \hat{L}_{S_{mn}}(\hat{f}) - \hat{L}_{S_{mn}}(f^*) \right] \tag{35}$$

$$\leq \mathop{\mathbb{E}}_{S_{mn}} \left[ L_E(\hat{f}) - \hat{L}_{S_{mn}}(\hat{f}) \right] \tag{36}$$

$$\leq 2 \mathop{\mathbb{E}}_{P_{1:n}} \left[ \mathcal{R}_{P_{1:n}^m}(\mathcal{F}) \right] + 2\mathcal{R}_{E^n}(\mathcal{F}), \tag{37}$$

where the last inequality is due to Theorem 2 $\qquad\square$

## D  PROOF OF THEOREM 4

We begin with a Theorem bounding the minimax excess risk in terms of Rademacher complexity for the case of i.i.d. learning. This Theorem is a generalisation of result from Sridharan (2011) that enables the use of the zero–one classification loss, instead of an absolute difference regression loss.

**Theorem 9.** *Let $\mathcal{F}$ be a class of models producing predictions in $\{-1, 1\}$, and let $\ell$ be the zero–one loss. We have that the minimax excess risk for problems in $\mathcal{P}$ is bounded from below by*

$$M(\mathcal{F}, \mathcal{P}, m) \geq \sup_{S_{2m}} \frac{1}{2} \hat{\mathcal{R}}_{S_{2m}}(\mathcal{F}) - \sup_{S_m} \frac{1}{4} \hat{\mathcal{R}}_{S_m}(\mathcal{F}).$$

*Proof.* Recall that the minimax excess risk is given by,

$$M(\mathcal{F}, \mathcal{P}, m) = \inf_A \sup_{P \in \mathcal{P}} \left\{ \mathop{\mathbb{E}}_{S_m} \left[ \mathop{\mathbb{E}}_{(\mathrm{x,y})} [\mathbf{1}(\mathrm{y}A(S_m)(\mathbf{x}) < 0)] \right] - \inf_{f \in \mathcal{F}} \mathop{\mathbb{E}}_{(\mathrm{x,y})} [\mathbf{1}(\mathrm{y}f(\mathbf{x}) < 0)] \right\}. \tag{38}$$

When models in $\mathcal{F}$ produce values in $\{-1, 1\}$, we note that the zero–one loss can be rewritten as

$$\ell_{01}(\hat{y}, y) = \mathbf{1}(y\hat{y} < 0) \tag{39}$$
$$= 1 - \mathbf{1}(y\hat{y} \geq 0) \tag{40}$$
$$= 1 - \frac{y\hat{y} + 1}{2}. \tag{41}$$

The minimax excess risk can therefore be expressed as

$$M(\mathcal{F}, \mathcal{P}, m) = \inf_A \sup_{P \in \mathcal{P}} \left\{ \sup_{f \in \mathcal{F}} \mathop{\mathbb{E}}_{(\mathbf{x,y})} [\mathrm{y}f(\mathbf{x})] - \mathop{\mathbb{E}}_{S_m} \left[ \mathop{\mathbb{E}}_{(\mathbf{x,y})} [\mathrm{y}A(S_m)(\mathbf{x})] \right] \right\}, \tag{42}$$

which can be bounded from below by

$$M(\mathcal{F}, \mathcal{P}, m) \geq \inf_A \sup_{\boldsymbol{x}_1,...,\boldsymbol{x}_{2m}} \sup_{y_1,...,y_{2m}} \left\{ \sup_{f \in \mathcal{F}} \frac{1}{4m} \sum_{i=1}^{2m} y_i f(\boldsymbol{x}_i) - \mathop{\mathbb{E}}_{S_m} \left[ \frac{1}{4m} \sum_{i=1}^{2m} y_i A(S_m)(\boldsymbol{x}_i) \right] \right\}, \tag{43}$$

where elements in $S_m$ are distributed according to the uniform distribution over the $(\boldsymbol{x}_i, y_i)$ pairs. We further observe

$$M(\mathcal{F}, \mathcal{P}, m) \geq \inf_A \sup_{\boldsymbol{x}_1,...,\boldsymbol{x}_{2m}} \mathop{\mathbb{E}}_{y_i \sim R} \left[ \sup_{f \in \mathcal{F}} \frac{1}{4m} \sum_{i=1}^{2m} \mathrm{y}_i f(\boldsymbol{x}_i) - \mathop{\mathbb{E}}_{S_m} \left[ \frac{1}{4m} \sum_{i=1}^{2m} \mathrm{y}_i A(S_m)(\boldsymbol{x}_i) \right] \right] \tag{44}$$

$$\geq \sup_{\boldsymbol{x}_1,...,\boldsymbol{x}_{2m}} \mathop{\mathbb{E}}_{y_i \sim R} \left[ \sup_{f \in \mathcal{F}} \frac{1}{4m} \sum_{i=1}^{2m} \mathrm{y}_i f(\boldsymbol{x}_i) \right] \tag{45}$$

$$- \sup_A \sup_{\boldsymbol{x}_1,...,\boldsymbol{x}_{2m}} \mathop{\mathbb{E}}_{y_i \sim R} \left[ \mathop{\mathbb{E}}_{S_m \sim P_U^m} \left[ \frac{1}{4m} \sum_{i=1}^{2m} \mathrm{y}_i A(S_m)(\boldsymbol{x}_i) \right] \right], \tag{46}$$

where the first inequality arises because we replace the supremum taken over $y_1, ..., y_{2m} \in \{-1, 1\}$ with an expectation over Rademacher random variables, $y_i \sim R$. The second inequality comes from splitting the infimum and supremum across two terms and subsequently simplifying. The first term is just a worst-case Rademacher complexity, so

$$M(\mathcal{F}, \mathcal{P}, m) \geq \sup_{S_{2m}} \frac{1}{2} \hat{\mathcal{R}}_{S_{2m}}(\mathcal{F}) - \sup_A \sup_{\boldsymbol{x}_1, ..., \boldsymbol{x}_{2m}} \mathbb{E}_{y_i \sim R} \left[ \mathbb{E}_{S_m} \left[ \frac{1}{2m} \sum_{i=1}^{4m} y_i A(S_m)(\boldsymbol{x}_i) \right] \right]. \quad (47)$$

Denote by $I \subset \mathbb{N}^{2m}$ the set of indices determining the $\boldsymbol{x}_i$ in $S_m$. The second term can be rewritten as

$$\sup_A \sup_{\boldsymbol{x}_1, ..., \boldsymbol{x}_{2m}} \mathbb{E}_{I \sim U[2m]^m} \left[ \mathbb{E}_{y_i \sim R} \left[ \frac{1}{4m} \sum_{i \in I} y_i A(S_m)(\boldsymbol{x}_i) \right] \right], \quad (48)$$

because the terms in the Rademacher sum corresponding to samples that do not appear in $S_m$ will evaluate to zero. This can then be bounded from above by

$$\sup_A \sup_{\boldsymbol{x}_1, ..., \boldsymbol{x}_{2m}} \mathbb{E}_{I \sim U[2m]^m} \left[ \mathbb{E}_{y_i \sim R} \left[ \frac{1}{4m} \sum_{i \in I} y_i A(S_m)(\boldsymbol{x}_i) \right] \right] \leq \sup_{S_m} \mathbb{E}_{y_i \sim R} \left[ \sup_{f \in \mathcal{F}} \frac{1}{4m} \sum_{\boldsymbol{x} \in S_m} y_i f(\boldsymbol{x}) \right]$$
$$(49)$$

$$= \sup_{S_m} \frac{1}{4} \hat{\mathcal{R}}_{S_m}(\mathcal{F}), \quad (50)$$

which concludes the proof. $\qquad \square$

The key insight for proving Theorem 4 is encapsulated in the following lemma.

**Lemma 1.** *For $\gamma \in [0, 1]$ we have*

$$M(\mathcal{F}, \mathcal{E}, mn) \geq \gamma M(\mathcal{F}, \mathcal{P}, mn) + (1 - \gamma) M(\mathcal{F}, \mathcal{P}, n).$$

*Proof.* Let $\mathcal{E}_1$ be the subset of $\mathcal{E}$ consisting of all environments containing only one domain each. Also, let $\mathcal{E}_2$ be the subset of $\mathcal{E}$ consisting of environments where each domain is a point mass. Let $\gamma_1 = \gamma$ and $\gamma_2 = 1 - \gamma$. The minimax excess risk can be lower bounded by

$$M(\mathcal{F}, \mathcal{E}, mn) \geq \inf_A \sum_{k=1}^2 \gamma_k \left( \sup_{E \in \mathcal{E}_k} \left\{ \mathbb{E}_{S_{mn}} [L_E^{01}(A(S_{mn})] - \inf_{f \in \mathcal{F}} L_E^{01}(f) \right\} \right) \quad (51)$$

$$\geq \sum_{k=1}^2 \gamma_k \left( \inf_A \sup_{E \in \mathcal{E}_k} \left\{ \mathbb{E}_{S_{mn}} [L_E^{01}(A(S_{mn})] - \inf_{f \in \mathcal{F}} L_E^{01}(f) \right\} \right) \quad (52)$$

$$= \gamma M(\mathcal{F}, \mathcal{P}, mn) + (1 - \gamma) \left( \inf_A \sup_{E \in \mathcal{E}_2} \left\{ \mathbb{E}_{S_{mn}} [L_E^{01}(A(S_{mn})] - \inf_{f \in \mathcal{F}} L_E^{01}(f) \right\} \right), \quad (53)$$

where the first inequality comes from the supremum of a set being lower bounded by supremum of a subset, the second inequality from a standard property of infima, and the final equality from the definition of $\mathcal{E}_1$.

We now turn our attention to the second term. Define the function,

$$C_m(S_n) = \cup_{i=1}^m S_n \quad (54)$$

Note that, because each $P \sim E \in \mathcal{E}$ is a point mass, we have that for every $S_{mn}$ there exists and $S_n$ such that $S_{mn} = C_m(S_n)$. Therefore, for every learning algorithm, $A$, that maps from sets of size $mn$ to models, there exists an algorithm $\tilde{A} = A \circ C_m$ that maps from the corresponding set of size $n$ to the same model. Therefore,

$$\inf_A \sup_{E \in \mathcal{E}_2} \left\{ \mathbb{E}_{S_{mn}} [L_E^{01}(A(S_{mn})] - \inf_{f \in \mathcal{F}} L_E^{01}(f) \right\} = \inf_A \sup_{P \in \mathcal{P}} \left\{ \mathbb{E}_{S_n} [L_P^{01}(A(S_n)] - \inf_{f \in \mathcal{F}} L_P^{01}(f) \right\} \quad (55)$$

$$= M(\mathcal{F}, \mathcal{P}, n), \quad (56)$$

from which the result follows. $\qquad \square$

We now prove Theorem 4.

**Theorem 4.** *Let $\mathcal{F}$ be a class of models producing predictions in $\{-1, 1\}$ and let $\ell$ be the zero–one loss. We have that the minimax excess risk for problems in $\mathcal{E}$ is bounded from below by*

$$M(\mathcal{F}, \mathcal{E}, mn) \geq \frac{\sup_{S_{2mn}} \hat{\mathcal{R}}_{S_{2mn}}(\mathcal{F}) + \sup_{S_{2n}} \hat{\mathcal{R}}_{S_{2n}}(\mathcal{F})}{4} - \frac{\sup_{S_{mn}} \hat{\mathcal{R}}_{S_{mn}}(\mathcal{F}) + \sup_{S_n} \hat{\mathcal{R}}_{S_n}(\mathcal{F})}{8}.$$

*Proof.* This is an immediate consequence of Theorem 9 and Lemma 1 with $\gamma = \frac{1}{2}$. We note that a tighter lower bound can be obtained with a different choice of $\gamma$, but this choice enables easier comparison with the upper bound on the excess risk of ERM. $\square$

# E    PROOF OF THEOREM 5

**Theorem 5.** *Let $\ell$ be a loss function taking values in $[0, 1]$. For a hypothesis class, $\mathcal{F}$, and any $\mathcal{G}$ such that $\ell \circ \mathcal{F} \subseteq \mathcal{G}$, then we have for any $E \in \mathcal{E}_{\mathcal{G},\sigma}$*

$$\mathbb{E}_{S_{mn}} \left[ \sup_{f \in \mathcal{F}} L_E(f) - \hat{L}_{S_{mn}}(f) \right] \leq 2 \mathbb{E}_{P_{1:n}} [\mathcal{R}_{P_{1:n}^m}(\ell \circ \mathcal{F})] + \sigma.$$

*Moreover, we have with confidence at least $1 - \delta$ over the realisations of $S_{mn}$, for all $f \in \mathcal{F}$, that*

$$L_E(f) \leq \hat{L}_{S_{mn}}(f) + 2\hat{\mathcal{R}}_{S_{mn}}(\ell \circ \mathcal{F}) + \sigma + 3\sqrt{\frac{\ln 2/\delta}{2mn}}.$$

*Proof.* First note that

$$\mathbb{E}_{S_{mn}} \left[ \sup_{f \in \mathcal{F}} L_E(f) - \hat{L}_{S_{mn}}(f) \right] = \mathbb{E}_{S_{mn}} \left[ \sup_{f \in \mathcal{F}} L_E(f) - \frac{1}{n} \sum_{j=1}^{n} L_{P_j}(f) + \frac{1}{n} \sum_{j=1}^{n} L_{P_j}(f) - \hat{L}_{S_{mn}}(f) \right] \tag{57}$$

$$\leq \mathbb{E}_{P_{1:n}} \left[ \sup_{f \in \mathcal{F}} L_E(f) - \frac{1}{n} \sum_{j=1}^{n} L_{P_j}(f) \right] + 2\mathcal{R}_{P_{1:n}^m}(\mathcal{F}), \tag{58}$$

from Theorem 1. Further, we have

$$\mathbb{E}_{P_{1:n}} \left[ \sup_{f \in \mathcal{F}} L_E(f) - \frac{1}{n} \sum_{j=1}^{n} L_{P_j}(f) \right] \leq \mathbb{E}_{P_{1:n}} \mathbb{E}_{P'_{1:n}} \left[ \sup_{f \in \mathcal{F}} \frac{1}{n} \sum_{j=1}^{n} (L_{P'_j}(f) - L_{P_j}(f)) \right] \tag{59}$$

$$\leq \mathbb{E}_{P_{1:n}} \mathbb{E}_{P'_{1:n}} \left[ \frac{1}{n} \sum_{j=1}^{n} \left( \sup_{f \in \mathcal{F}} L_{P'_j}(f) - L_{P_j}(f) \right) \right], \tag{60}$$

$$\leq \sigma, \tag{61}$$

which yields the first part of the result. The second part of the result follows from the same line of reasoning as used in Theorem 2. $\square$

# F    PROOF OF THEOREM 6

**Theorem 6.** *Let $\mathcal{F}$ be a class of models producing predictions in $\{-1, 1\}$ and $\ell$ be the zero–one loss. For any $\mathcal{G}$ such that $d_{\mathcal{G}}$ takes values in $[0, 1]$, we have that the minimax excess risk for problems in $\mathcal{E}_{\mathcal{G},\sigma}$ is bounded from below by*

$$M(\mathcal{F}, \mathcal{E}_{\mathcal{G},\sigma}, mn) \geq \frac{\sup_{S_{2mn}} \hat{\mathcal{R}}_{S_{2mn}}(\mathcal{F}) + \sup_{S_{2n}} \sigma\hat{\mathcal{R}}_{S_{2n}}(\mathcal{F})}{4} - \frac{\sup_{S_{mn}} \hat{\mathcal{R}}_{S_{mn}}(\mathcal{F}) + \sup_{S_n} \sigma\hat{\mathcal{R}}_{S_n}(\mathcal{F})}{8}.$$

*Proof.* For any $E' \in \mathcal{E}$ and $P_0 \in \mathcal{P}$, we define $P \sim E$ as

$$P = (1 - \sigma)P_0 + \sigma P', \tag{62}$$

where $P' \sim E'$. Observe that $E \in \mathcal{E}_{\mathcal{G}, \sigma}$,

$$d_{\mathcal{G}}(P_1, P_2) = \sup_{g \in \mathcal{G}} \left| \mathbb{E}_{(\mathbf{x}, \mathbf{y}) \sim P_1} [g(\mathbf{x}, \mathbf{y})] - \mathbb{E}_{(\mathbf{x}, \mathbf{y}) \sim P_2} [g(\mathbf{x}, \mathbf{y})] \right| \tag{63}$$

$$= \sup_{g \in \mathcal{G}} \left| \int_{X \times Y} g(\mathbf{x}, \mathbf{y}) \cdot ((1 - \sigma)dP_0 + \sigma dP_1') - \int_{X \times Y} g(\mathbf{x}, \mathbf{y}) \cdot ((1 - \sigma)dP_0 + \sigma dP_2') \right| \tag{64}$$

$$= \sup_{g \in \mathcal{G}} \left| \int_{X \times Y} g(\mathbf{x}, \mathbf{y}) \cdot \sigma(dP_1' - dP_2') \right| \tag{65}$$

$$= \sigma \sup_{g \in \mathcal{G}} \left| \mathbb{E}_{(\mathbf{x}, \mathbf{y}) \sim P_1'} [g(\mathbf{x}, \mathbf{y})] - \mathbb{E}_{(\mathbf{x}, \mathbf{y}) \sim P_2'} [g(\mathbf{x}, \mathbf{y})] \right| \tag{66}$$

$$\leq \sigma. \tag{67}$$

Let $\mathcal{E}_1$ be the set of all DG problems that contain only a single domain, and define

$$\mathcal{E}_2 = \{ E \,|\, E' \in \mathcal{E}, \, P_0 \in \mathcal{P} \}. \tag{68}$$

Then we apply the same decomposition used in the proof for Lemma 1,

$$M(\mathcal{F}, \mathcal{E}_{\mathcal{G}, \sigma}, mn) \geq \inf_A \frac{1}{2} \sum_{k=1}^{2} \left( \sup_{E \in \mathcal{E}_k} \left\{ \mathbb{E}_{S_{mn}} [L_E^{01}(A(S_{mn}))] - \inf_{f \in \mathcal{F}} L_E^{01}(f) \right\} \right) \tag{69}$$

$$\geq \frac{1}{2} \sum_{k=1}^{2} \left( \inf_A \sup_{E \in \mathcal{E}_k} \left\{ \mathbb{E}_{S_{mn}} [L_E^{01}(A(S_{mn}))] - \inf_{f \in \mathcal{F}} L_E^{01}(f) \right\} \right) \tag{70}$$

$$= \frac{1}{2} M(\mathcal{F}, \mathcal{P}, mn) + \frac{1}{2} \left( \inf_A \sup_{E \in \mathcal{E}_2} \left\{ \mathbb{E}_{S_{mn}} [L_E^{01}(A(S_{mn}))] - \inf_{f \in \mathcal{F}} L_E^{01}(f) \right\} \right). \tag{71}$$

From the definition of $\mathcal{E}_2$, we can rewrite the second term as

$$\inf_A \sup_{E', P_0} \left\{ \mathbb{E}_{S_{mn}} [\sigma L_{E'}(A(S_{mn})) + (1 - \sigma) L_{P_0}(A(S_{mn}))] \right. \tag{72}$$

$$\left. - \inf_{f \in \mathcal{F}} (\sigma L_{E'}(f) + (1 - \sigma) L_{P_0}(f)) \right\} \tag{73}$$

$$\geq \inf_A \sup_{E', P_0} \left\{ \mathbb{E}_{P_{1:n} \sim E'^n} \mathbb{E}_{S_{mn} \sim P_{1:n}^m} [\sigma L_{E'}(A(S_{mn}))] + \mathbb{E}_{S_{mn} \sim P_0^{mn}} [(1 - \sigma) L_{P_0}(A(S_{mn}))] \right. \tag{74}$$

$$\left. - \inf_{f \in \mathcal{F}} (\sigma L_{E'}(f) + (1 - \sigma) L_{P_0}(f)) \right\} \tag{75}$$

$$\tag{76}$$

We proceed by considering two cases. First, assume the optimal model for the worst-case DG problem, $E'$, does not have a zero–one error of 1. Choose $P_0$ to be a point mass on one of the points classified correctly by the optimal model. Then the optimal model for $E'$ is also optimal for $P_0$ and achieves zero error. Noting also that the expected loss of any algorithm on $P_0$ is greater than or equal zero, we can further lower bound the above by

$$\sigma \inf_A \sup_{E' \in \mathcal{E}} \left\{ \mathbb{E}_{S_{mn}} [L_{E'}(A(S_{mn}))] - \inf_f L_{E'}(f) \right\}. \tag{77}$$

In the case where the optimal model for $E'$ does classify every point incorrectly, choose $P_0$ to be any point that can be generated by a domain in supp $E'$. The expected loss of any algorithm and

the loss of the optimal model are both one, and therefore cancel, yielding the same result as the first case. The main result follows from noting

$$\sigma \inf_{A} \sup_{E' \in \mathcal{E}} \left\{ \mathbb{E}_{S_{mn}} \left[ L_{E'}(A(S_{mn})) \right] - \inf_{f} L_{E'}(f) \right\} = \sigma M(\mathcal{F}, \mathcal{E}, mn), \tag{78}$$

and subsequently applying Lemma 1 with $\gamma = 0$. □

# G   PROOF OF THEOREM 7

**Theorem 7.** *Let $\ell$ be a loss function taking values in $[0,1]$. For a hypothesis class, $\mathcal{F}$, and any $E \in \mathcal{E}_\rho$, we have that*

$$\mathbb{E}_{S_{mn}} \left[ L_E(\hat{f}) \right] - L_E(f^*) \leq 2\rho \, \mathbb{E}_{P_{1:n}} \left[ \mathcal{R}_{P_{1:n}^m}(\mathcal{F}) \right] + (\rho - 1) \, \mathbb{E}_{S_{mn}} \left[ \hat{L}_{S_{mn}}(\hat{f}) \right].$$

*Proof.* Follow the proof of Theorem 3 until

$$\mathbb{E}_{S_{mn}} \left[ L_E(\hat{f}) \right] - L_E(f^*) \leq 2\rho \, \mathbb{E}_{P_{1:n}} \left[ \mathcal{R}_{P_{1:n}^m}(\ell \circ \mathcal{F}) \right] + (\rho - 1) \, \mathbb{E}_{S_{mn}} \left[ \hat{L}_{S_{mn}}(\hat{f}) \right]. \tag{79}$$

The result follows from applying a change of measure and applying the definition of $\mathcal{E}_\rho$,

$$\mathbb{E}_{S_{mn}} \left[ L_E(\hat{f}) - \hat{L}_{S_{mn}}(\hat{f}) \right] \leq \mathbb{E}_{P_{1:n}} \mathbb{E}_{S_{mn}} \left[ \frac{\rho}{n} \sum_{j=1}^{n} L_{P_j}(\hat{f}) - \hat{L}_{S_{mn}}(\hat{f}) \right] \tag{80}$$

$$= \mathbb{E}_{P_{1:n}} \mathbb{E}_{S_{mn}} \left[ \frac{\rho}{n} \sum_{j=1}^{n} L_{P_j}(\hat{f}) - \rho \hat{L}_{S_{mn}}(\hat{f}) + \rho \hat{L}_{S_{mn}}(\hat{f}) - \hat{L}_{S_{mn}}(\hat{f}) \right] \tag{81}$$

$$\leq 2\rho \, \mathbb{E}_{P_{1:n}} \left[ \mathcal{R}_{P_{1:n}^m}(\ell \circ \mathcal{F}) \right] + (\rho - 1) \, \mathbb{E}_{S_{mn}} \left[ \hat{L}_{S_{mn}}(\hat{f}) \right]. \tag{82}$$

□

# H   PROOF OF THEOREM 8

**Theorem 8.** *Let $\mathcal{F}$ be a class of models producing predictions in $\{-1, 1\}$ and $\ell$ be the zero–one loss. We have that the minimax risk for problem in $\mathcal{E}_\rho$ is bounded from below by*

$$M(\mathcal{F}, \mathcal{E}_\rho, mn) \geq \sup_{S_{2mn}} \frac{1}{2} \hat{\mathcal{R}}_{S_{2mn}}(\mathcal{F}) - \frac{1}{4} \sup_{S_{mn}} \hat{\mathcal{R}}(\mathcal{F}).$$

*Proof.* Notice that $\mathcal{E}_1$ is the set of all DG problems containing a single domain. From the definition of $\mathcal{E}_\rho$ we have $\mathcal{E}_1 \subseteq \mathcal{E}_\rho$. So,

$$M(\mathcal{F}, \mathcal{E}_\rho, mn) \geq M(\mathcal{F}, \mathcal{E}_1, mn) \tag{83}$$
$$= M(\mathcal{F}, \mathcal{P}, mn). \tag{84}$$

The result follows by applying Theorem 9. □

# I   EXPERIMENTAL ILLUSTRATION

We conduct small-scale experiments to illustrate the practical impact of our theoretical results. More specifically, we focus on a modification of the well-known MNIST hand-written digit classification task. We embed the original $28 \times 28$ images into a larger $64 \times 64$ grid, where each unique placement corresponds to a separate domain. In total there are 1,369 possible ways to place the digit, and from these we select uniformly at random 1,000 domains for training, 169 for validation, and 200 for testing. Additionally, each domain has an associated unique mixture component of rotating the

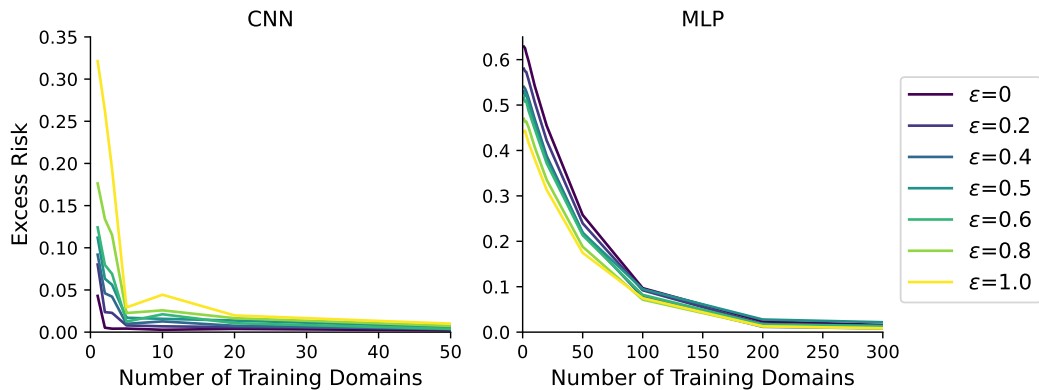

Figure 3: The CNN model includes a suitable invariance that helps it generalize across domains better than the MLP model.

digit, which is used with a probability $\epsilon$. The selected rotations are between $-45°$ and $45°$. In our experiments we vary the number of training domains to measure its impact on the generalization to new unseen domains. We control the total training set size to be the same across all numbers of domains, and we achieve this by associating each example in the dataset with a unique domain, with balancing the number of examples assigned to each domain. We repeat each experiment with three random seeds and calculate the average.

We use two models: a small Convolutional Neural Network (CNN) with global average pooling, and a Multi-Layer Perceptron (MLP). The purpose of these is to show that having a hypothesis class that includes suitable invariances leads to better excess risk. The CNN model consists of two convolutional layers followed by one dense layer, with global average pooling before the dense layer. The MLP model consists of three dense layers. Both models use ReLU non-linearity activation between the layers. We train each model for 10 epochs using Adam optimizer with learning rate of 0.001, cross-entropy loss and training minibatch of 64 items. We use early stopping via checkpoint selection, where we evaluate the model on the validation set after each epoch.

Translations and rotations of images can both be represented at permutations of the pixels, so these transforms can therefore can be modelled as a finite group, $G$. The CNN is invariant to the position of the digit, due to the global average pooling layer, but will does not explicitly encode rotation invariance. As such, the class of CNNs we consider is $(G, \epsilon)$-invariant, because we only apply a rotation with probability $\epsilon$. In contrast, the MLP does not encode either of these invariances, so we can expect it will need to see a large number of training domains before being able to approach the optimal within-class risk.

We show the results in Figure 3, comparing the two models with different rotation probabilities, $\epsilon$. When $\epsilon$ is small, very few training domains are needed in order to minimise the excess risk, because the differences in domains is mainly explained by translations. As $\epsilon$ grows, the CNN requires more training domains to converge towards the best possible model for this corresponding DG problem. However, the MLP converges towards the optimal model considerably slower, and at the same rate for all values of $\epsilon$, because it does not have any of the correct built-in invariances.

