# OpenReview forum: "On the Limitations of General Purpose Domain Generalisation Methods"
_ICLR.cc/2025/Conference — Submitted to ICLR 2025_

### Official Review · Reviewer_Lbjm · 2024-10-30

**Soundness:** 2
**Presentation:** 3
**Contribution:** 1
**Rating:** 3
**Confidence:** 1

**Summary:**

This paper investigates the problem of DG methods fail to substantially outperform ERM on rigorous DG benchmarks from a learning theory viewpoint. Particularly, the authors show that all learning algorithms converge slowly w.r.t. the number of training domains. For the constrained settings, ERM can achieve optimal learning rate towards the oracle. Overall, the current theoretical findings do not capture the fundamental aspects of DG problem, with conclusions that appear overly assertive.

**Strengths:**

- Present theoretical analysis on based on the **Rademacher** complexity measure.
- Lower bound on the minimax excess risk is given.

**Weaknesses:**

- The authors show that all learning algorithm converge slowly with respect to the number of training domains, implying lack of algorithmic progress.  Specifically, they  assert universal  ERM optimality under asymptotic conditions with finite domains and samples, potentially overlooking domain-specific invariance properties and structural assumptions. Such a claim is too assertive (A typical example is the CMNIST,  where ERM  consistently performs worse than well-motivated DG algorithms).  Without any specific characterizations on distribution shift [1]，it is unlikely to capture the key factors governing   generalization performance across domains.

    [1] **An Information-theoretic Approach to Distribution Shifts, NeurIPS 2021.**

- The theoretical results do not involve the algorithm architecture. For modern DNNs, the **Rademacher** complexity bounds are basically vacuous, we cannot gain meaningful insights from the current claims.
- In practical, the i.i.d. assumption of domain is too restrictive to apply. Besides, as shown in previous study [2,3],  proper assumptions about the problem statement and data-generating process plays a key role in characterize DG.

    [2] **On learning invariant representations for domain adaptation, ICML 2019.**

    [3] **The risks of invariant risk minimization, ICLR 2021.**

- The main results are basically rooted in the pioneering theoretical investigation on ERM [4, 5]. The technical contributions on theoretical prove are quite limited.

    [4] **Rademacher and Gaussian complexities: Risk bounds and structural results. *JMLR*, 2002**

    [5]  **Foundations of Machine Learning, MIT Press, 2018.**

**Questions:**

Please refer to weakness

---

### Official Review · Reviewer_wWQ8 · 2024-10-31

**Soundness:** 1
**Presentation:** 3
**Contribution:** 2
**Rating:** 3
**Confidence:** 5

**Summary:**

This paper conducts generalization analysis for the domain generalization problem under the Rademacher complexity framework. The authors derive upper bounds for the domain generalization error and excess risk, and lower bounds for the minimax excess risk. They further extend the results under the bounded Integral Probability Metrics and bounded density ratio settings. Based on their main theoretical results, the authors claim that ERM better approximates the optimal model and no model can substantially outperform ERM under general purpose DG settings.

**Strengths:**

The paper is generally well-written and easy to follow. Multiple new generalization bounds for the domain generalization problem are derived, which to some extent explain the competitive performance of ERM on DG benchmarks e.g. DomainBed. Several insights are also given based on the theoretical results that can help to improve the performance of ERM under some DG settings.

**Weaknesses:**

Given the current results presented by the authors, I have to disagree with the main claims of the paper that no model substantially outperforms ERM. Here are the reasons:

* The problem formulation of DG deviates from the mainstream settings. This paper focuses on minimizing the expected population risk of the model on randomly sampled target domains (eq. 5). This objective is equivalent to ERM on i.i.d sampled domains, but is already shown to be vulnerable to OOD data. Instead, DG is usually formalized as a minimax optimization problem [1] e.g. $\min_f \max_P L_P(f)$, which is not covered in this paper. This results in a different optimal model $f^*$.

* It is consensus that Rademacher complexity is not sufficient to explain generalization of deep learning. To let $\mathcal{R}_m(\mathcal{F})$ scale with $1/\sqrt{m}$, one require $m$ to be significantly larger than the capacity of the models, which is hardly met for modern deep networks. Instead, they can easily fit the given random labeled datasets and we will get $\mathcal{R}_m(\mathcal{F}) \approx 1$ in most cases which invalidates the bounds.

* In the definition of minimax excess risk, $\mathcal{P}$ is considered to be the set of all possible sample distributions over $X \times Y$. This is rarely the case in practice since $X$ and $Y$ always share some common causal relationships among different domains. The authors claim that $\mathcal{P}$ can be easily extended to a certain set of distributions, but to my understanding the proof of Theorem 4 relies on this condition. At (eq. 42 $\to$ eq.43), the labels $y_1$ to $y_{2m}$ can no longer be arbitrarily selected when $\mathcal{P}$ has changed. Thus, the lower bounds may not apply to practical learning settings.

* There is strong evidence that ERM results in poor generalization under certain conditions, e.g. [2]. I do not see conflicts between the settings of [2] and this paper. Explanations on why the results in this paper do not apply to Colored MNIST in [2] should be included.

[1] Sagawa S, Koh PW, Hashimoto TB, Liang P. Distributionally Robust Neural Networks. InInternational Conference on Learning Representations 2019 Sep 25.
[2] Arjovsky M, Bottou L, Gulrajani I, Lopez-Paz D. Invariant risk minimization. arXiv preprint arXiv:1907.02893. 2019 Jul 5.

**Questions:**

* Please discuss how will the result change under the minimax formulation.
* Please discuss how Theorem 4 will change when $\mathcal{P}$ is altered.
* Please explain the counter example in [2].

---

### Official Review · Reviewer_HXRV · 2024-11-01

**Soundness:** 3
**Presentation:** 3
**Contribution:** 2
**Rating:** 5
**Confidence:** 4

**Summary:**

This paper presents a theoretical analysis related to the challenges of domain generalization (DG) methods that can reliably perform across varied data distributions. Despite numerous empirical efforts, existing DG methods consistently struggle to outperform empirical risk minimization (ERM). The authors uses tools from empirical process theory and minimax theory to provide upper and lower bounds on the excess risk depending on the number of domains, number of samples per domains and some properties on the distribution of the domains.

**Strengths:**

This paper presents some theoretical results that quantify the difficulty of generalizing to some unseen domain.

**Weaknesses:**

The setup of this paper is quite broad, with very general theorems, making it challenging to assess the precise difficulty of the domain generalization problem. One popular approach in domain generalization is invariant risk minimization (IRM), introduced by Arjovsky et al., and subsequently explored in other studies. This line of research aims to identify patterns that are consistent across multiple domains, thereby disregarding spurious correlations that vary across domains. This approach aligns with (and is motivated by) causal inference, which assumes that only the causal components of correlations remain stable across domains.


In this paper, the authors begin with a general setup (Section 3), assuming that distributions are generated i.i.d. from a distribution over the space of distributions. This aligns well with the Bayesian framework, where such a setup is known as a hierarchical prior. However, in the context of domain generalization, if all domains are generated independently and identically, it becomes unclear what form of similarity the model is expected to learn across domains. If there is no notion of similarity we expect to learn from domains, what is the motivation behind borrowing information from various domains? The theorems in Section 3 are fairly basic, applying standard symmetrization techniques, which leads to general but uninformative convergence rates. For instance, the derived rate, $1/\sqrt{mn} + 1/\sqrt{n}$, essentially simplifies to $1/\sqrt{n}$ (as $m > 1$), meaning that the rate depends primarily on the number of domains. This result is unsurprising, given the independence assumed across all domains.

In Section 4, the authors introduce conditions that allow for potentially faster rates, with these conditions depending on the distance between domains. However, it remains unclear whether the upper and lower bounds align precisely or if the dependence on $\sigma$ in the upper bound is optimal.

Additionally, the current presentation of the results may be challenging for readers less familiar with empirical process theory. I suggest assuming a specific complexity measure for the underlying function class (such as covering number) and then expressing the results in terms of $(m, n, \sigma)$ for both the upper and lower bounds.

**Questions:**

1. Suppose we assume that the best predictor $E[Y | X]$ remains the same across all the domains. Can we estimate the predictor at a better rate?

2. Does the upper bound in Theorem 5 match the lower bound in Theorem 6?

3. Can we have some understanding/simplification of the result when we have covariate shift (i.e. distribution of $X$ changes but the distribution of $Y|X$ remains same)?

4. Could we interpret the $\sigma$ parameter as representing the variance of the distribution over domains ($E$), rather than using an IPM? For instance, if $E$ has a very low variance, this would imply that the domains are highly similar, which should lead to an improved rate of convergence. Is this perspective something that can be derived from the paper’s methodology?

---

### Official Review · Reviewer_4uA1 · 2024-11-03

**Soundness:** 2
**Presentation:** 2
**Contribution:** 2
**Rating:** 5
**Confidence:** 3

**Summary:**

This paper considers the domain generalization problem, where a model trained on some domain is required to generalize to new domains.  This paper derives upper bounds on the excess risk of ERM, as long as minimax lower bounds for the excess risk. Three settings with different restrictions on the difference of the domains are considered. For all three settings, the authors demonstrate that ERM has an optimal rate of convergence towards the best possible model.

**Strengths:**

This paper considers the important problem of domain generalization, which is pervasive in modern machine learning. The authors give an interpretation of why ERM performs well under domain generalization setup, which matches the observation in reality.

**Weaknesses:**

My main concern is that some of this papers' claim is not rigorous. For example, in the discussion of Theorem 4 (equation (11)), the authors claim that the minimax excess risk scales as $\Omega(\frac{1}{\sqrt{mn}} + \frac{1}{\sqrt{n}})$, since the two terms in lower bound are both $\Omega(\frac{1}{\sqrt{mn}} + \frac{1}{\sqrt{n}})$, and $\Omega(\frac{1}{\sqrt{mn}} + \frac{1}{\sqrt{n}}) - \Omega(\frac{1}{\sqrt{mn}} + \frac{1}{\sqrt{n}})$ scales as $\Omega(\frac{1}{\sqrt{mn}} + \frac{1}{\sqrt{n}})$. This claim is not true in general.

Also, the authors claim that the risk "scales as" something, where many related quantities are hiding in the "$\Omega$". Then they claim that this best-case behaviour matches the worst-case scaling behaviour of ERM, so that ERM already has the optimal scaling behaviour for general DG problems. However, since many things are hiding in the "$\Omega$", I do not think this claim is valuable enough.

**Questions:**

My questions are mainly related to the weaknesses.
1. Can you provide further explanation of why $\Omega(\frac{1}{\sqrt{mn}} + \frac{1}{\sqrt{n}}) - \Omega(\frac{1}{\sqrt{mn}} + \frac{1}{\sqrt{n}})$ scales as $\Omega(\frac{1}{\sqrt{mn}} + \frac{1}{\sqrt{n}})$ is true?
2. What terms are hiding in $\Omega$?

---

### Official Review · Reviewer_Zb4q · 2024-11-09

**Soundness:** 3
**Presentation:** 3
**Contribution:** 2
**Rating:** 3
**Confidence:** 4

**Summary:**

The paper considers upper bounds for ERM, as well as min-max lower bounds for domain generalization, in settings in which we are given n total environments and m samples per environment. It considers a "fully worst-case" setting in which case no additional structure exists between environments; it also includes "somewhat benign" settings in which some notion of similarity between the environments is bounded (e.g. an IPM distance between the covariate distributions, or a bounded density ration between the covariate distributions). The main claim is that a Rademacher complexity-based upper bound of ERM matches (to leading order) the min-max lower bounds in these settings --- suggesting ERM is the "optimal" thing one can hope for.

**Strengths:**

The paper compiles several structural assumptions of "similarity" between the domains (overlap and IPM distance), and works out the upper and lower bounds on what is statistically achievable in both settings.

**Weaknesses:**

On a technical front, the paper is fairly straightforward: since suprema in the Rademacher complexity nicely "split" by triangle inequality, the upper bounds boil down to two application of the standard Rademacher bound machinery. The min-max lower bounds also follow from fairly standard machinery and are fairly expected.

This of course would be not a problem if the results were surprising or suggested interesting training interventions --- which unfortunately I don't think it's the case here:
(1) The main reason for a lot of the developments mentioned in the related works and introduction (e.g. the causal modeling view in IRM) was introduced as a richer way to model distribution shift. In fact, in the a lot of the causality-inspired works (e.g. IRM), the goal is not to only do well *in expectation* over environments, but *in the worst case* over some set of environments (e.g. all possible sets of interventions). Because the paper commits to a metric of performance "in expectation" over the choice of environment, and takes a worst-case view over the set of predictors in some class --- the fact that the Rademacher complexity characterizes both the performance of ERM, and the information-theoretic optimum is not surprising.
(2) The "actionable insights" suggested in the paper (amount of regularization to pick) are also predicated upon the "in-expectation" view of generalizing over a domain distribution. The more typical viewpoint in recent DG work is to focus on a "worst-case" notion of generalization in some set of domains (e.g. all interventions on a subset of variables in a causal graph) --- in which case choosing the regularization strength is much trickier. In fact, as Gulrajani-Lopez Paz show --- a "cheating way" to choose regularization strength in IRM, if allowed to "peek" in the test set, is why a lot of papers over-reported the performance of IRM.

Overall, I think neither the technical ingredients in the proofs, nor the results themselves stand well on their own.

**Questions:**

--

---

### Meta-Review · Area_Chair_3vi2 · 2024-12-26

**Metareview:**

This paper investigates domain generalization (DG) from a learning theory perspective, focusing on the performance of ERM. The authors provide upper and lower bounds for the excess risk in DG settings, considering different assumptions about domain similarity, and conclude that ERM is optimal under certain conditions. While the paper provides useful theoretical insights into domain generalization and ERM, the overall contribution and novelty of the paper are questioned across the reviews.

**Additional Comments On Reviewer Discussion:**

The main issue across the reviews is the limited novelty and practical relevance of the results. The technical approach, while sound, applies well-known machinery (such as Rademacher complexity and symmetrization) and leads to results that are seen as fairly expected or unsurprising. The claim that ERM is optimal in domain generalization, particularly when the domains are independent and identically distributed (i.i.d.), is viewed as overly assertive. Many reviewers point out that this does not account for key domain-specific invariance properties or structural assumptions that are often critical in DG tasks. Additionally, the i.i.d. assumption itself is seen as too restrictive for real-world applications.

Several reviewers also critique the lack of practical insights, as the findings do not suggest novel training interventions or algorithmic improvements. The paper's focus on "in-expectation" generalization and failure to consider "worst-case" generalization scenarios (which are more common in recent DG work) further weakens its contribution.

---

### Decision · Program_Chairs · 2025-01-22

Reject